# Awareness and Attitudes Regarding Industrial Food Fortification in Mongolia and Harbin

**DOI:** 10.3390/nu11010201

**Published:** 2019-01-19

**Authors:** Sabri Bromage, Enkhmaa Gonchigsumlaa, Margaret Traeger, Bayarbat Magsar, Qifan Wang, Jorick Bater, Hewei Li, Davaasambuu Ganmaa

**Affiliations:** 1Department of Nutrition, Harvard T.H. Chan School of Public Health, Boston, MA 02115, USA; 2Department of Monitoring, State Agency for Specialized Inspection, Ulaanbaatar 15170, Mongolia; uchir0319@yahoo.com; 3Department of Sociology, Yale University, New Haven, CT 06511, USA; margaret.traeger@yale.edu; 4Department of Graduate Studies, Heilongjiang University of Chinese Medicine, Harbin 150040, China; bayarbatmagsar@gmail.com (B.M.); lihewei6872@163.com (H.L.); 5Department of Humanities and Management, Heilongjiang University of Chinese Medicine, Harbin 150040, China; qifanwang24@gmail.com; 6Department of Global Health and Population, Harvard T.H. Chan School of Public Health, Boston, MA 02115, USA; bater@hsph.harvard.edu; 7Department of Nutrition, Harvard T.H. Chan School of Public Health and Channing Division of Network Medicine, Department of Medicine, Brigham and Women’s Hospital and Harvard Medical School, Boston, MA 02115, USA; gdavaasa@hsph.harvard.edu

**Keywords:** food fortification, nutrition policy, consumer attitudes, food choice, nutrition knowledge, functional food, Mongolia, China

## Abstract

This study assessed awareness and attitudes regarding industrial food fortification among adults in urban and rural Mongolia, and the city of Harbin, China. Between 2014 and 2017, surveys were collected from healthy men and women aged ≥18 years (182 Harbin residents and 129 urban and rural Mongolians participating in a nationwide nutrition survey in Mongolia). Survey reproducibility was assessed among 69 Mongolian participants to whom it was administered twice (summer and winter). Findings revealed that only 19% of rural and 30% of urban Mongolians, and 48% of Harbin residents were aware that industrial fortification is practiced in their countries. For most food groups evaluated, at least half of Mongolians and less than half of Harbin residents thought fortification was government-mandated (only the addition of iodine with salt is actually mandated in both countries). Fifty-five percent of rural and urban Mongolians favored mandatory fortification of foods, 14% disapproved of it, and 31% were uncertain (compared with 25%, 38%, and 37% respectively in Harbin). Upon learning that the primary purpose of adding vitamin D to milk is to prevent rickets, 75% of Mongolians but only 18% of Harbin residents favored mandatory fortification, while 42% of Harbin residents favored voluntary fortification (compared with <10% of Mongolians). In conclusion, in Mongolia and Harbin, awareness and understanding of food fortification is low, as is receptivity toward mandatory fortification. Health promotion and social marketing should be designed to create an enabling environment for increasing supply and demand of fortified foods, in support of upcoming program implementation in Mongolia and potential future legislation in northeern China.

## 1. Introduction

The prevalence of multiple micronutrient deficiencies in China and Mongolia has decreased substantially in recent decades. Yet, in spite of overall improvements, deficiencies persist, particularly among poor and rural areas of both countries [1,2]. Analysis of the 2011 China Health and Nutrition Survey indicated dietary inadequacies of calcium, iron, zinc, selenium, vitamin A, thiamine, riboflavin, and vitamin C to be extremely common among boys and girls of different age groups between 4–17 years [3]. Biochemical analyses of Chinese children have found a high prevalence of calcium, iron, zinc, and selenium deficiency; high prevalence of mild iodine deficiency, marginal vitamin A deficiency, and vitamin D insufficiency; and moderate prevalence of thiamine, riboflavin, and vitamin B12 deficiency nationwide [4]. In a Mongolian nationwide survey from 2012 to 2016, dietary inadequacies of thiamin, folate, and vitamins A, D, and E were common among Mongolian adults [5], while a 2006 study of young children aged 6–24 months noted dietary inadequacies of calcium, iron, zinc, and vitamins A and C to be of concern [6]. Biochemical deficiencies of iron, zinc, folic acid, and vitamins A and D have been reported among Mongolian women of reproductive age and young children [2,7,8].

A range of strategies may be considered to address population micronutrient deficiencies, among which industrial food fortification is considered to be a relatively broad impact, cost effective, and sustainable one, particularly when mandated by law [9]. However, advocacy for national fortification is challenged by the high level of commitment required from government agencies and food producers, extensive data required for effective program design and monitoring, and financial and infrastructure requirements for implementation [10,11]. At present, the only example of mandatory fortification in both China and Mongolia is that of salt with iodine, which has been effective in reducing the prevalence of iodine deficiency in both countries since its implementation in 1995 [12,13]. Voluntary fortification is practiced in both countries and is especially widespread in China [14,15,16]. In China, voluntarily iron-fortified soy sauce is particularly important and has been effective in reducing the prevalence of anemia since its approval in 2002 and its expansion during ten years of public education, social marketing, and infrastructure development in cooperation with the Global Alliance for Improved Nutrition [17,18]. Other examples of fortified foods in China are baby milk formula, nutrient- and probiotic-fortified milk and dairy products, and micronutrient-fortified biscuits, breads, and breakfast cereals [14,15].

Mongolia’s small population and centralized production of staple foods should facilitate expanded fortification legislation there [5]. Eleven years after the conclusion of an Asian Development Bank-funded infrastructural project and effectiveness trial [7,19], renewed public sector advocacy efforts led the Mongolian parliament to pass a law in April 2018 expanding mandatory industrial fortification beyond that of salt; national program implementation is planned in January 2020 [20]. At the time of this writing, drafted guidance for this program includes mandatory fortification of wheat flour with specified concentration of eight vitamins and mineral fortificants. By contrast, efforts to legislate mandatory fortification in China, which have been underway for decades, are not explicitly described in the government’s current long-term nutrition plan for 2017–2030 [21]. Despite high production and consumption of staple foods [22], and encouraging results from recent community trials [23,24,25] and modeling studies [26,27], national fortification legislation has been challenged by wide variation in dietary and production patterns across China’s enormous, complex, and decentralized food system, the lack of consumer demand, and lack of political interest [28,29,30].

In addition to top-down lobbying aimed at garnering legislative, ministerial, and industry support, public understanding of the role of fortification is necessary to create an enabling political and commercial environment, induce supply of fortified foods, and ensure that fortified foods are consumed [9]. As part of a collaboration between researchers working in Mongolia and Harbin, the present study assessed and compared public awareness of, and attitudes toward, food fortification among adults living in rural and urban Mongolia, and Harbin, the capital of China’s northernmost province (Heilongjiang) which neighbors Mongolia, in order to identify priorities for social marketing and health promotion campaigns in support of effective program implementation in Mongolia and potential future fortification legislation and updated guidelines in northern China.

## 2. Materials and Methods

Mongolian participants were drawn from a subset of adults participating in a nationwide dietary survey who were also administered the fortification survey [5]. Rural Mongolians were randomly sampled from the Orkhon and Bayandalai soums (districts) of Bulgan and Omnogobi aimags (provinces), respectively, while urban participants were drawn from the provincial capitals of Bulgan and Omnogobi (the cities of Bulgan and Dalanzadgad, respectively) and from central and peri-urban areas of Ulaanbaatar (the capital of Mongolia). Harbin participants were recruited at random from public areas in Harbin, and from among the office staff and service workers at Harbin Institute of Technology and Heilongjiang University of Chinese Medicine. As part of their participation in the larger dietary survey, Mongolian participants received a monetary incentive equivalent to approximately 7 USD. The study’s methodology received approval from the Ethical Review Boards of the Mongolian Ministry of Health and First Affiliated Hospital of Heilongjiang University of Chinese Medicine, and the Harvard Longwood Medical Area Institutional Review Board.

Survey questions were adapted from methodology used to assess awareness, attitudes, and behaviors towards fortified foods [31]. The survey instrument is provided in Appendix A. The instrument captured awareness and attitudes toward the practice of food fortification generally and with respect to specific foods; awareness and attitudes toward mandatory and voluntary fortification; preferences and behaviors related to the purchase and consumption of fortified foods by oneself and one’s children; awareness and utilization of fortification-related information from different sources; and contextualizing information regarding food consumption and purchasing behaviors as well as individual and household socio-demographics. The complete survey was translated, back-translated, and piloted among research assistants assigned to data collection at each of the three study sites (rural Mongolia, urban Mongolia, and Harbin) and adjusted as appropriate to ensure interpretability and usefulness of the questions as well as their comparability across study sites. Participants provided informed consent prior to data collection. Forty-two rural and 87 urban Mongolian participants completed the survey from December 2014 to January 2015. A subset of 41 rural and 28 urban Mongolians also completed the survey in the preceding summer of 2014; these 69 individuals provided two surveys each for the purpose of assessing survey reproducibility. One-hundred and eighty-two Harbin participants completed the survey from December 2016 to January 2017. At each site, surveys were self-administered with a trained research assistant nearby to clarify questions as needed in a neutral manner and ensure completeness of the responses.

The frequency and proportion of different survey responses were tabulated for each of the three study sites. Fisher’s exact test was used to assess the statistical significance of differences observed in the distribution of responses to each question across study sites. Logistic and multinomial logistic regression models were run to identify associations between responses to selected survey questions and study site (rural/urban Mongolia and Harbin) adjusted for sex, age in years and the square of age, university education, self-reported bodyweight classification, and proportion of food shopping done for the household, as well as to identify the independent effects of these covariates across sites (in this analysis, some survey responses were condensed to ensure stability of parameter estimates for study site). An additional set of models was run to understand the role of fortification awareness and attitudes on food consumption decisions across study sites by predicting the reported proclivity for purchasing or consuming specific fortified foods (Appendix A) using responses to other survey questions, adjusting for study site and socio-demographic and lifestyle characteristics.

Reproducibility of selected survey questions was assessed by calculating percent agreement between paired summer and winter responses within the subsample of 28 rural and 41 urban Mongolians who completed the survey twice. For each survey question, a two-sample test of proportions was used to assess significant differences in percent agreement in responses between rural and urban Mongolia, and to assess whether percent agreement significantly exceeded that which would be expected by chance. In both rural and urban Mongolia, reproducibility of the majority of survey questions (in terms of percent agreement between paired winter and summer responses) was greater than that expected by chance alone, and for all but one question (which assessed awareness of fortification overall), reproducibility did not differ significantly between rural and urban participants (Appendix A). With the exception of one survey question that asked about the purpose of milk fortification, the survey did not explicitly educate participants, and while there is a possibility that participants’ general awareness of fortification was affected by exposure to the survey, we assume the effect of this exposure on survey reproducibility six months later to be small.

## 3. Results

Rural Mongolian participants were significantly older than their urban counterparts and Harbin participants (median age: 42, 32, and 37 years, respectively), and less educated (% of participants holding a university degree: 25%, 82%, 87%, respectively) (Table 1). Compared with rural and urban Mongolians, Harbin residents consumed more fruits and vegetables (% consuming 2+ servings of fruit/day: 11%, 23%, 29%, respectively; vegetables: 49%, 48%, 78%), while rural Mongolians drank the most milk (% consuming at least 500 mL/day: 36%, 21%, 28%, respectively). Most participants considered themselves to be “the right weight” (57%, 57%, and 56% of rural Mongolians, urban Mongolians, and Harbin residents, respectively) or overweight (40%, 36%, 30%).

A significantly higher proportion of rural Mongolians were unsure as to whether vitamins or minerals are sometimes added to foods and drinks in comparison with their urban counterparts and Harbin residents (64%, 49%, 40%, respectively), while only 19%, 30%, and 48% knew that micronutrients are in fact added and 17%, 21%, and 12% responded that they are not (Table 2). Across the three study sites (urban Mongolia, rural Mongolia, and Harbin), 55%, 44%, and 42% of participants were unsure if industrial fortification is mandated by the government, respectively, while only 26%, 27%, and 23% were aware that it is and 19%, 39%, and 37% answered it is not. When asked whether specific foods were subject to mandatory fortification, the distribution of responses (“yes”, “no”, “unsure”) did not differ significantly between urban and rural Mongolians for most foods (except milk and margarine), while for most foods, at least 50% of urban and rural Mongolians believed fortification was mandatory. By contrast, less than half of Harbin residents believed that fortification was mandatory for most foods (except breakfast cereals and milk, for which 63% and 51% of participants responded that fortification was mandatory). In each study site, the two most common means by which participants reported becoming aware of food fortification were food packaging (specifically, the ingredient list) and advertisements in multimedia; the ingredient list was also the most common place on food packaging that participants reported looking at when trying to determine whether foods had been fortified.

Over 90% of rural and urban Mongolians and only 52% of Harbin residents reported that they would prefer their children drank vitamin D-fortified milk, and 79% of rural and 87% of urban Mongolians thought that other Mongolians would be receptive to milk fortification compared with only 65% of Harbin residents who thought similarly (Table 3). Upon discovering that a particular food was fortified, rural and urban Mongolians were similarly influenced with respect to their likelihood of consuming it (40% and 44% would be more likely to, 2% and 6% would be less likely to, and 31% and 40% thought it would depend on the food or micronutrient in question), while Harbin residents were more divided (only 19% would be more likely to consume it, 25% would be less likely, and 40% responded that it would depend). In both rural and urban Mongolia, 55% of participants favored mandatory fortification, 14% disapproved of it, and 31% were unsure, while only 25%, 38%, and 37% of Harbin residents were for, against, or unsure, respectively. Upon being informed of the primary purpose of vitamin D fortification of milk, more Harbin residents favored voluntary (42%) rather than mandatory (18%) fortification, while 76% of rural and 75% of urban Mongolians favored mandatory fortification. For most food groups, as was the case with awareness of fortification, rural and urban Mongolians were generally more similar to one another than with Harbin residents with respect to their proclivity to purchase fortified foods based on the fact that they were fortified.

Adjusting for selected socio-demographic and lifestyle characteristics, no significant differences in the odds of any survey responses were observed between rural and urban Mongolians (Appendix A), and compared with residents of Harbin, urban and rural Mongolians had significantly lower adjusted odds of fortification awareness; were more partial to consumption of vitamin D fortified milk by their children, more likely to believe that their fellow nationals would be receptive to fortified milk, more likely to purchase foods that they knew to be fortified, and more supportive of mandatory fortification. Across study sites, those with a university education had significantly higher adjusted odds of awareness that fortification of certain foods is mandatory (OR = 3.304, *p* = 0.0194), were more likely to agree with mandatory vitamin D fortification of milk (OR = 2.684, *p* = 0.0335) and mandatory fortification in general (OR = 2.204, *p* = 0.0464), and to believe others would be receptive to milk fortification (OR = 2.073, *p* = 0.0423). Adjusted models also showed that, across study sites, females were significantly more willing to pay 5%–10% more for fortified foods (OR = 2.102, *p* = 0.0105), those who purchased half or more of the food for their household were more likely to be aware of fortification (OR = 3.001, p = 0.0085), and those who perceived themselves as underweight were less likely to be aware of fortification (OR = 0.490, *p* = 0.0171) or respond that vitamin D fortification of milk should be mandatory (OR = 0.430, *p* = 0.0142). Age was not independently associated with participants’ responses to any survey questions.

Survey questions were varyingly predictive of reported proclivity to consume specific fortified foods across study sites in adjusted models (Appendix A), the most predictive question of which was whether one knows that a food is in fact fortified (this question was associated with reported consumption of five of ten fortified foods). Two questions (whether milk should be fortified with vitamin D given that its purpose is to reduce risk of bone deformities in children, and whether the government mandates fortification of any foods) were each independently associated with consumption of three fortified foods. Of the ten foods analyzed, consumption of noodles and salt, and to a lesser extent flavored or smart water, margarine-like spreads, and flour were the most strongly associated with fortification awareness and attitudes (in terms of the number of assessed questions with which consumption of those foods was associated).

## 4. Discussion

The present study demonstrates a low level of awareness and understanding about industrial food fortification and national fortification policy among rural and urban Mongolians, and residents of Harbin, China. In comparison with residents of Harbin, Mongolians were generally less aware that fortification is practiced in their respective country (when in fact salt fortification is mandatory, and voluntary fortification is becoming more common for other food vehicles), but were also more likely to believe that fortification of specific foods is mandated by the government (when in fact, with the exception of mandatory salt fortification, it is not). Mongolians were also more likely than residents of Harbin to prefer that manufacturers fortify foods and that fortification be mandatory. Between rural and urban Mongolians, rural Mongolians were generally less knowledgeable of fortification and fortification law, but did not differ significantly from their urban counterparts with respect to their receptiveness toward fortification.

To our knowledge, the present study is the first to compare public awareness and attitudes about industrial food fortification in Mongolia and China, although studies have been conducted within each country. Salt is an instructive example because it is the only food subject to mandatory fortification in both countries. Among 2220 pregnant women in the 2016 Fifth Mongolian National Nutrition Survey, 85.4% were aware of iodized salt, awareness was similar between urban (86.4%) and rural areas (84.1%), and only 0.5% did not know what type of salt (iodized or non-iodized) they used [2], while among 298 pregnant urban Chinese women, 76.9% were aware of fortification, and a “low” percentage were “very familiar” [32]. (Interestingly, the present study showed only 30% of urban Mongolians and 48% of Harbin residents were aware that micronutrients are added to any foods and drinks at all, which may reflect the fact that it is easier to recollect information about specific foods rather than all foods, as the former requires the individual to think more carefully). Awareness of salt fortification among Ulaanbaatar pregnant women in the 2017 National Nutrition Survey (86.9%) is less than what it was among parents (92%) and post-partum women (98%) in a smaller survey of 838 Ulaanbaatar residents in 1996, the year after fortification was initiated [33]. In 1996, multimedia and written materials (rather than information received in person) were primary sources of information about salt iodization in Ulaanbaatar; based on the present study 20 years later, urban Mongolians still receive such information primarily from these sources. The present study also found that only 59% of urban Mongolians were certain that salt fortification is mandatory, indicating a discrepancy between knowledge that salt is fortified (86.9%, according to the 2017 National Nutrition Survey) and knowledge of the policy behind it.

Despite being generally more aware and knowledgeable about the practice of industrial fortification than urban Mongolians, fewer Harbin participants in the present study were aware that salt fortification is mandatory (33% of Harbin residents vs. 59% of urban Mongolians), significantly fewer consumed salt because it was fortified (17% vs. 41%), fewer were receptive to the idea of mandatory fortification (25% vs. 55%), and fewer would prefer that their children drink fortified milk (52% vs. 96%). These differences may in part reflect different cultural attitudes toward the role of government in regulating the food supply, which are in turn reflected by the extent of voluntary fortification and supplementation in China. While voluntary fortification and adult micronutrient supplementation are uncommon in Mongolia [2,5], the Chinese food market is saturated with voluntarily-fortified foods [14,15] which provide 10.1% of calcium, 4.0% of iron, and 4.0% of zinc to pregnant women in Beijing, and supplements provide more than twice these contributions [32]. Similarly, compliance with home fortification and supplementation, and effectiveness of education and social marketing interventions in China are high [17,34,35,36,37,38,39,40], while receptiveness toward environmental approaches (e.g., fortification and biofortification) is lower [41,42,43]. Mandatory industrial fortification has generally enjoyed less political support in China than Mongolia in part due to a history of outdated legislation that has not supported effective regulation [28] and the fact that while the Mongolian diet is relatively homogenous [5] across the country, staple grain consumption differs significantly between Chinese provinces [44] and urban vs. rural localities [45], and the penetrability of a uniform fortification program would vary accordingly (in Harbin, for example, wheat flour fortification should be more effective, given the relative prominence of wheat flour in the diet of northeastern China). This does not necessarily mean that mandatory fortification should be discounted, but that a less centralized policy (which allows or accounts for variable subnational penetrability of industrial fortification) may be more prudent. However, implementing such a policy in the context of China’s complex food system—in which administrative boundaries, relative consumption of potential vehicles, and producers’ target markets may be difficult to map, do not neatly overlap, and are dynamic—is a challenging prospect. Furthermore, unlike in rural Mongolia, where flour production is highly centralized [46], much of the cereal consumed in rural China is processed in small mills [47] and cannot be easily fortified. In the specific case of wheat flour fortification with folic acid, it has also been argued that fortification may be less appropriate than supplementation given China’s relatively high rate of planned pregnancies [48]; it has alternately been argued that dietary folate consumption and the prevalence of preconception supplementation may not adequately reduce the incidence of birth defects in the absence of fortification [49,50].

In addition to their different policy environments, differences in attitudes toward food fortification between Harbin and Mongolia may also relate to more basic differences in the distribution of consumer and product characteristics, and other determinants that mediate willingness to purchase and consume fortified foods. After adjusting for study site and demographic variables, the present study found that respondents with a university education and those who purchased at least half of the food for their household were more aware of fortification across sites, that university education was also associated with a preference toward fortification (and mandatory vs. voluntary fortification), and that females were more willing to pay more for fortified food. Prior studies have found that more educated and female consumers are generally more likely to consume functional foods, although it is a question as to how consistently these findings may be generalized across countries [51]. Interestingly, self-classification as underweight was independently associated with lower fortification awareness and lower approval of milk fortification with vitamin D; while these statistics may be subject to sources unmeasured and confounding, they may suggest that those who perceive themselves as malnourished in these populations prioritize factors other than fortification (such as the quantity or intrinsic nutritional value of foods) when making consumption decisions.

Recent analysis by Jahn and colleagues determined three major categories of drivers—positive attitude toward fortification, awareness of population nutrient deficiency, and perceived appropriateness of fortification for a given product—as requisites in consumers’ decisions to purchase vitamin D-fortified foods [52]. In the present study, we explored a comparable analysis, albeit using different available predictors emphasizing the domains assessed in the present survey (i.e., awareness and attitudes toward fortification). In this analysis, which simultaneously controlled for fortification awareness and attitudes, knowledge that a given food had in fact been fortified was independently associated with proclivity to purchase or consume five of ten assessed foods across study sites, and in all cases, consumers were either more likely to consume foods they knew to be fortified or were more willing to consume them depending on the fortification vehicle or micronutrient in question. Primarily, this analysis suggests that the influence of fortification awareness may outweigh personal attitudes toward fortification in determining one’s consumption of fortified foods in these populations. This finding is concerning given the low level of awareness observed in each of the study sites, and warrants further investigation to understand potential interactions and mediating effects between fortification awareness and attitudes in these populations.

Research on cross-country samples indicates that while determinants of functional food use are broadly consistent across countries and only modestly associated with local cultural values, the local contribution and interpretation of these determinants naturally vary according to international differences in population demographics, psychological factors, and availability and manifestation of functional foods [53,54,55,56]. In the present study, crude differences between rural and urban Mongolians in their willingness to pay for fortified foods and preference toward vitamin D fortification of milk were not evident following multivariable adjustment, suggesting that determinants of fortification attitudes are similar across urban and rural Mongolia. Conversely, crude differences between Mongolians and residents of Harbin were generally robust to adjustment; while these differences may in part be related to intrinsic cultural values, it is also probable that a more detailed survey designed to understand more complex variation in psychosocial and behavioral variation could identify shared dimensions of awareness and attitudes across the two countries. To our knowledge, published studies have not compared fortification awareness and attitudes between Asian populations despite the proliferation of fortified and other functional foods in Asia (prior cross-country studies and the majority of research concerned with drivers of functional food consumption have mostly focused on Europe). The purpose of the present analysis is primarily descriptive and intended as a first step, after which future studies should delve more deeply to identify demographic, psychological, and food-specific drivers of fortified food use in and across Asian populations.

## 5. Conclusions

The low level of public awareness and understanding of fortification in Mongolia and Harbin, as indicated in the present study, is concerning. Increasing public knowledge through health promotion, education, and social marketing will increase supply and demand of voluntarily-fortified foods that more effectively address population deficiencies in both countries, garner political will for exploring legislation options and regulation in China, and ease the implementation of upcoming mandatory fortification in Mongolia. The generalizability of our findings is limited by our small sample size, as is our exploration of how fortification awareness and attitudes are associated with socio-demographic characteristics of individuals within study sites. Generalizability of our findings to China may also be limited beyond Harbin or the surrounding Heilongjiang province. Future public awareness activities in China and Mongolia would therefore benefit from more extensive and detailed baseline assessments to allow more specific targeting of population subgroups.

Despite circumscription of Chinese participants to Harbin, we find this study’s comparison between Harbin and Mongolia to be convenient in light of their proximity with one another, and interpretable given that these areas’ dietary patterns share enough resemblance with one another to allow use of a uniform survey instrument (the Chinese province whose diet is most similar to that of Mongolia is Inner Mongolia, which would also provide an interesting comparison, but one from which results would be less generalizable across northern China given Inner Mongolia’s relative socio-ethnic uniqueness within China). We also find the comparison between China and Mongolia to be instructive given the marked differences we have discussed in these countries’ food systems, policy environments, and cultures. Variation between China and Mongolia (and between closely co-located countries in Asia overall) provides a useful lens with which to study relationships between macro-level variables and drivers of fortified and functional food use, and to identify cross-cultural drivers in the region. This will be an important area of investigation in the near future as industrial and bio-fortification become more prevalent in Asia. Comparing these drivers to those identified in Europe and the Americas would also further contribute to global research and policy guidance in fortification.

## Figures and Tables

**Table 1 nutrients-11-00201-t001:** Characteristics of study population.

Survey Question	Rural Mongolia (*n* = 42)		Urban Mongolia (*n* = 87)		Harbin (*n* = 182)	
	*n*	%	*n*	%	*n*	%
Sex						
Male	22	52	42	48	76	42
Female	20	48	45	52	106	58
Age Group ^a,b^						
18–29	1	2	31	35	46	24
30–39	14	29	24	27	51	27
40–49	13	27	15	17	37	20
50–59	10	21	13	15	36	19
60+	10	21	5	6	19	10
Education Level ^a,b^						
No university degree	27	75	15	18	23	13
University degree	9	25	69	82	155	87
How much of the food shopping do you usually do for your household? ^a,b,c^						
None	14	34	9	11	20	11
Less than half	3	7	18	21	76	42
About half	9	22	23	27	42	23
All or most	15	37	35	41	43	24
How many servings of fruit do you typically eat in a day? ^b,c^						
<1	5	14	10	13	4	2
1	28	76	52	65	61	34
2	3	8	14	18	64	35
3+	1	2	4	4	53	29
How many servings of vegetables do you typically eat in a day? ^a,b,c^						
<1	4	10	0	0	4	2
1	16	41	45	52	36	20
2	14	36	23	27	68	38
3+	5	13	18	21	73	40
How much milk do you typically drink in a day? ^a,b^						
≤250 mL	25	64	59	79	130	73
500 mL	8	21	13	17	34	19
750 mL	0		2	3	12	7
≥1 L	6	15	1	1	3	2
Do you consider yourself the right weight, underweight or overweight? ^b^						
The right weight	24	57	47	57	98	56
Overweight	17	40	30	36	52	30
Underweight	1	2	6	7	26	15

Footnote: Percentages are calculated after excluding missing values. ^a,b,c^ denote that response distributions differ significantly between ^a^ rural and urban Mongolia, ^b^ rural Mongolia and Harbin, and ^c^ urban Mongolia and Harbin (Fisher’s exact test *p* < 0.05).

**Table 2 nutrients-11-00201-t002:** Awareness of food fortification.

Survey Question	Rural Mongolia (*n* = 42)		Urban Mongolia (*n* = 87)		Harbin (*n* = 182)	
	*n*	%	*n*	%	*n*	%
Are vitamins or minerals sometimes added to foods and drinks by the manufacturer? ^b,c^						
Yes	8	19	26	30	88	48
No	7	17	18	21	22	12
Unsure	27	64	43	49	72	40
Does the government mandate the addition of vitamins or minerals to certain foods?						
Yes	11	26	15	17	42	23
No	8	19	34	39	68	37
Unsure	23	55	38	44	72	40
Is it mandatory that vitamins or minerals be added to the following? (Yes/No/Unsure)						
Flour ^b,c^	36/0/6	86/0/14	69/4/14	79/5/16	66/59/49	38/34/28
Breakfast cereal ^b,c^	23/2/17	55/5/40	49/8/30	56/9/34	109/33/32	63/19/18
Milk ^b,c^	33/2/7	79/5/17	56/5/26	64/6/30	89/46/39	51/26/22
Margarine/similar spread ^a,b,c^	20/4/18	48/10/43	61/9/17	70/10/20	32/75/67	18/43/39
Salt ^b,c^	28/4/10	67/10/24	51/16/20	59/18/23	58/70/46	33/40/26
Fruit juice ^b,c^	30/3/9	71/7/21	60/10/17	69/11/20	86/63/25	49/36/14
Candy ^b,c^	16/9/17	38/21/40	29/21/37	33/24/43	46/85/43	26/49/25
Last time you bought or consumed a food with added vitamins or minerals, how did you know it was fortified? *						
Read it on the package ^a,c^	16	38	62	73	54	30
Multimedia advertisement ^c^	14	33	22	26	71	39
Read it in an article	2	5	7	8	28	15
Word of mouth ^c^	6	14	13	15	51	28
Other source	0	0	2	2	17	9
Don’t know ^a,b^	9	21	7	8	11	6
Have not bought such foods	1	2	2	2	12	7
Where on food packaging have you seen a statement regarding added vitamins or minerals? *						
A statement on the front	12	29	34	39	80	44
Nutrition information panel ^a,b^	7	17	32	37	82	45
Ingredient list	16	38	37	43	91	50
Elsewhere on the package ^c^	1	2	3	3	24	13
Don’t know ^b^	8	19	6	7	9	5
Have not seen such statements	3	7	2	2	8	4
If you wanted to know if a food had added vitamins or minerals, where would you look? *						
A statement on the front ^c^	15	36	23	27	82	45
Nutrition information panel ^a,b,c^	4	10	26	30	83	46
Ingredient list	22	52	46	53	88	48
Elsewhere on the package	2	5	2	2	12	7
Don’t know	6	14	7	8	15	8

Footnote: Percentages are calculated after excluding missing values. ^a,b,c^ denote that response distributions differ significantly between ^a^ rural and urban Mongolia, ^b^ rural Mongolia and Harbin, and ^c^ urban Mongolia and Harbin (Fisher’s exact test *p* < 0.05). * Multiple responses are permitted for this question.

**Table 3 nutrients-11-00201-t003:** Attitudes toward food fortification.

Survey Question	Rural Mongolia (*n* = 42)		Urban Mongolia (*n* = 87)		Harbin (*n* = 182)	
	*n*	%	*n*	%	*n*	%
Would you prefer that your child(ren) drink vitamin D fortified milk or unfortified milk? ^b,c^						
Fortified	39	93	82	96	92	52
Unfortified	3	7	3	4	84	48
Do you think others would be receptive to fortification of milk products? ^c^						
Yes	33	79	76	87	119	65
No	8	19	10	11	60	33
Unsure	1	2	1	1	3	2
If you found that a food or drink had added vitamins or minerals, would this make you: ^b,c^						
More likely to buy or consume it	17	40	38	44	34	19
Less likely to buy or consume it	1	2	5	6	45	25
It makes no difference	7	17	3	3	26	14
Depends on the food, drink, vitamin, or mineral	13	31	35	40	72	40
Don’t know	4	10	6	7	5	3
How much more would you be willing to pay for fortified food products? ^a,c^						
I would not pay more	14	33	28	32	45	25
5%	10	24	29	33	52	29
10%	6	14	14	16	52	29
15%	10	24	5	6	27	15
20%	2	5	11	13	6	3
Should the government mandate the addition of vitamins and minerals to food? ^b,c^						
Yes	23	55	48	55	46	25
No	6	14	12	14	69	38
Don’t know	13	31	27	31	67	37
Vitamin D may be added to milk to reduce risk of bone deformities in children. Knowing this, would you say: ^a,b,c^						
Milk fortification should be mandatory	32	76	65	75	33	18
Milk fortification should be voluntary	0	0	9	10	76	42
Unsure	4	10	12	14	45	25
Don’t care	6	14	1	1	28	15
What is the strength of your opinion on your previous answer? ^b,c^						
Don’t know	2	5	0	0	8	4
Moderate	4	10	6	7	56	31
Somewhat strong	24	57	49	57	91	50
Very strong	12	29	31	36	27	15
Which of the following foods and drinks do you buy or consume because they have added vitamins or minerals? *						
Flour ^b,c^	25	60	36	41	18	10
Breakfast cereal ^b,c^	1	2	9	10	95	52
Noodles ^b,c^	15	36	24	28	24	13
Milk ^b,c^	1	2	5	6	43	24
Margarine/similar spreads ^a,b,c^	9	21	18	21	16	9
Other dairy products ^a,c^	18	43	62	71	89	49
Salt ^b,c^	21	50	41	47	31	17
Fruit juice ^c^	10	24	33	38	43	24
Flavored or smart water ^a,b,c^	1	2	9	10	43	24
Other foods	2	5	3	3	10	5

Footnote: Percentages are calculated after excluding missing values. ^a,b,c^ denote that response distributions differ significantly between ^a^ rural and urban Mongolia, ^b^ rural Mongolia and Harbin, and ^c^ urban Mongolia and Harbin (Fisher’s exact test *p* < 0.05). * Multiple responses are permitted for this question.

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
