# Peer review of "Awareness and Attitudes Regarding Industrial Food Fortification in Mongolia and Harbin"

_nutrients, 2019, doi:10.3390/nu11010201_

Round 1

Reviewer 1 Report

The authors have conducted a survey in Mongolia and Harbin to learn about awareness and attitudes regarding industrial food fortification. The paper is well written and addresses an important and timely topic. Despite merits, I have some concerns regarding the manuscript that I will detail below. The comments relate to the positioning and conceptual model/data analysis, which both have consequences for the intended contribution of the manuscript. I hope these comments will help the author(s) further develop their work.

(1) Although I can clearly see the impetus and convenience of researchers from two countries collaborating, I wonder how the specific selection of Mongolia and Harbin informs scholars interested in understanding food fortification acceptance. Is there any way the observed differences can be generalized to other areas or populations? I had the feeling that the results are of high relevance to policymakers in Mongolia and (Northern) China, but it is unclear what people not explicitly concerned with these regions can do with the results. I would like to see a rationale what the fields gains from this comparision, and how the observed differences can be linked to structural characteristics.

(2) Related to my first concern, I found it hard to understand the intended incremental contribution of the paper. Aside from the country focus, a bulk of research exists that has focused on acceptance of fortified food. Some of this work has more generally looked at reasons behind choices of fortified food (Urala & Lähteenmäki 2003) or changing attitudes over time (Urala & Lähteenmäki 2007), whereas others have explicitly examined the role of socio-demographic determinants in this regard (Ares & Gambaro 2007; Verbek 2005). There has even been a recent review paper on consumers' acceptance and preferences for nutrition-modified and functional dairy products (Bimbo et al. 2017). I would love to see integration of what we already know into this paper.

(3) My last comment is about the conceptual model, or lack thereof, paired with data analysis. The authors report a number of descriptive statistics but do not test any group differences. I think this is problematic, especially as there are differences in sample composition (e.g., age and education; p. 3). Would it be possible to conduct an ANCOVA or regression analysis that explcititly tests for the effect of region (rural Mongolia, urban Mongolia, and Harbin), while controlling for age, education, etc.? It might also be interesting to discuss the findings along those discussed by Jahn, Tsalis and Lähteenmäki (2019). These authors have proposed a model that links attitudes toward food fortification with personal benefit and perceptions of the need for fortification in society. In this vein, the study context (see my concern #1) could also be used for interpretation.

Overall, I really enjoyed reading and think the manuscript covers an important topic. Yet I felt the authors could elaborate more on their intended contribution to the literature on food fortification acceptance. I wish you good luck!

Ares, G., & Gambaro, A. (2007). Influence of gender, age and motives underlying food
choice on perceived healthiness and willingness to try functional foods. Appetite,
49(1), 148–158.

Bimbo, F., Bonanno, A., Nocella, G., Viscecchia, R., Nardone, G., De Devitiis, B., et al.
(2017). Consumers' acceptance and preferences for nutrition-modified and functional
dairy products: A systematic review. Appetite, 113, 141–154.

Jahn, S., Tsalis, G., & Lähteenmäki, L. (2019). How attitude towards food fortification can lead to purchase intention. Appetite, 133, 370–377.

Urala, N., & Lähteenmäki, L. (2003). Reasons behind consumers' functional food choices.
Nutrition & Food Science, 33(4), 148–158.

Urala, N., & Lähteenmäki, L. (2007). Consumers' changing attitudes towards functional
foods. Food Quality and Preference, 18(1), 1–12.

Verbeke, W. (2005). Consumer acceptance of functional foods: Socio-demographic, cognitive
and attitudinal determinants. Food Quality and Preference, 16(1), 45–57.

Reviewer 2 Report

Review for :  Awareness and attitudes regarding industrial food  fortification in Mongolia and Harbin

 Ln 21 “We assessed” Should you use 2nd person in research writing

Ln 45 What are the nutrients of concern for each population?

Ln 47 high coverage- odd wording

Ln 55 What are other examples of voluntary fortification

Ln 64 What will be fortified in Mongolia as a result of this legislation?

Materials and Methods

Any incentive to participants

Give more information for what the survey assesses and the scale as a summary in the methods

How was the survey it validated?

What is Cronbach Alpha of the survey

LN 104 So the survey was administered as an interview?- Clarify

Rural Mongolia participants are very different from the rest of your sample.

Not sure it is really necessary to present test-re-test throughout your result tables, Summarize as part of your methods.

For the participants in the winter -1st administration…were they also educated about fortification as they took the survey… wouldn’t this be a confounding factor for really comparing retest reliability?

Round 2

Reviewer 1 Report

I think the authors did a great job revising their paper. Although I would have loved to see a more extended discussion of what we already know about food fortification acceptance in the Introduction, the additions to the Discussion and Conclusion sections are valuable. I also found it interesting to see the difference between rural and urban Mongolia disappear after controlling for sociodemographics, while in general urban Mongolia resembles more closely the Harbin region. To conclude, I commend the authors for their fine revision.